META-RESEARCH ARTICLE

# Examining linguistic shifts between preprints and publications

**David N. Nicholson**[1], **Vincent Rubinetti**[1,2], **Dongbo Hu**[1], **Marvin Thielk**[3], **Lawrence E. Hunter**[4], **Casey S. Greene**[1,2,5] *

**1** Department of Systems Pharmacology and Translational Therapeutics, Perelman School of Medicine University of Pennsylvania, Philadelphia, Pennsylvania, United States of America, **2** Center for Health AI, University of Colorado School of Medicine, Aurora, Colorado, United States of America, **3** Elsevier, Philadelphia, Pennsylvania, United States of America, **4** Center for Computational Pharmacology, University of Colorado School of Medicine, Aurora, Colorado, United States of America, **5** Department of Biochemistry and Molecular Genetics, University of Colorado School of Medicine, Aurora, Colorado, United States of America

* greenescientist@gmail.com

**Data Availability Statement:** An online version of this manuscript is available under a Creative Commons Attribution License at https://greenelab.github.io/annorxiver_manuscript/. Source code for the research portions of this project is dual

## Abstract

Preprints allow researchers to make their findings available to the scientific community before they have undergone peer review. Studies on preprints within bioRxiv have been largely focused on article metadata and how often these preprints are downloaded, cited, published, and discussed online. A missing element that has yet to be examined is the language contained within the bioRxiv preprint repository. We sought to compare and contrast linguistic features within bioRxiv preprints to published biomedical text as a whole as this is an excellent opportunity to examine how peer review changes these documents. The most prevalent features that changed appear to be associated with typesetting and mentions of supporting information sections or additional files. In addition to text comparison, we created document embeddings derived from a preprint-trained word2vec model. We found that these embeddings are able to parse out different scientific approaches and concepts, link unannotated preprint–peer-reviewed article pairs, and identify journals that publish linguistically similar papers to a given preprint. We also used these embeddings to examine factors associated with the time elapsed between the posting of a first preprint and the appearance of a peer-reviewed publication. We found that preprints with more versions posted and more textual changes took longer to publish. Lastly, we constructed a web application (https://greenelab.github.io/preprint-similarity-search/) that allows users to identify which journals and articles that are most linguistically similar to a bioRxiv or medRxiv preprint as well as observe where the preprint would be positioned within a published article landscape.

## Introduction

The dissemination of research findings is key to science. Initially, much of this communication happened orally [1]. During the 17th century, the predominant form of communication shifted to personal letters shared from one scientist to another [1]. Scientific journals didn't

licensed under the BSD 3-Clause and Creative Commons Public Domain Dedication Licenses at https://github.com/greenelab/annorxiver. The preprint similarity search website can be found at https://greenelab.github.io/preprint-similarity-search/, and code for the website is available under a BSD-2-Clause Plus Patent License at https://github.com/greenelab/preprint-similarity-search. All corresponding data for every figure in this manuscript is available at https://github.com/greenelab/annorxiver/blob/master/FIGURE_DATA_SOURCE.md. Full text access for the bioRxiv repository is available at https://www.biorxiv.org/tdm. Access to PubMed Central's Open Access subset is available on NCBI's FTP server at https://www.ncbi.nlm.nih.gov/pmc/tools/ftp/. The New York Times Annotated Corpus (NYTAC) can be accessed from the Linguistic Data Consortium at https://catalog.ldc.upenn.edu/LDC2008T19 where there may be a $150 to $300 fee depending on membership status.

**Funding:** This work was supported by grants from the Gordon Betty Moore Foundation (GBMF4552) and the National Institutes of Health's National Human Genome Research Institute (NHGRI) under award R01 HG010067 to CSG and the National Institutes of Health's NHGRI under award T32 HG00046 to DNN. The funders had no role in study design, data collection and analysis, decision to publish, or preparation of the manuscript.

**Competing interests:** I have read the journal's policy and the authors of this manuscript have the following competing interests: Marvin Thielk receives a salary from Elsevier Inc. where he contributes NLP expertise to health content operations. Elsevier did not restrict the results or interpretations that could be published in this manuscript.

**Abbreviations:** API, application programming interface; CBOW, continuous bag of words; DOI, digital object identifier; KL, Kullback–Leibler; NIH/NLM, National Institute of Health's Library of Medicine; NYTAC, New York Times Annotated Corpus; PC, principal component; PCA, principal component analysis; PMC, PubMed Central; PMCOA, Pubmed Central's Open Access.

become a predominant mode of communication until the 19th and 20th centuries when the first journal was created [1–3]. Although scientific journals became the primary method of communication, they added high maintenance costs and long publication times to scientific discourse [2,3]. Some scientists' solutions to these issues have been to communicate through preprints, which are scholarly works that have yet to undergo peer review process [4,5].

Preprints are commonly hosted on online repositories, where users have open and easy access to these works. Notable repositories include arXiv [6], bioRxiv [7], and medRxiv [8]; however, there are over 60 different repositories available [9]. The burgeoning uptake of preprints in life sciences has been examined through research focused on metadata from the bioRxiv repository. For example, life science preprints are being posted at an increasing rate [10]. Furthermore, these preprints are being rapidly shared on social media, routinely downloaded, and cited [11]. Some preprint categories are shared on social media by both scientists and non-scientists [12]. About two-thirds to three-quarters of preprints are eventually published [13,14], and life science articles that have a corresponding preprint version are cited and discussed more often than articles without them [15–17]. Preprints take an average of 160 days to be published in the peer-reviewed literature [18], and those with multiple versions take longer to publish [18].

The rapid uptake of preprints in the life sciences also poses challenges. Preprint repositories receive a growing number of submissions [19]. Linking preprints with their published counterparts is vital to maintaining scholarly discourse consistency, but this task is challenging to perform manually [16,20,21]. Errors and omissions in linkage result in missing links and consequently erroneous metadata. Furthermore, repositories based on standard publishing tools are not designed to show how the textual content of preprints is altered due to the peer review process [19]. Certain scientists have expressed concern that competitors could scoop them by making results available before publication [19,22]. Preprint repositories by definition do not perform in-depth peer review, which can result in posted preprints containing inconsistent results or conclusions [17,20,23,24]; however, an analysis of preprints posted at the beginning of 2020 revealed that over 50% underwent minor changes in the abstract text as they were published, but over 70% did not change or only had simple rearrangements to panels and tables [25]. Despite a growing emphasis on using preprints to examine the publishing process within life sciences, how these findings relate to the text of all documents in bioRxiv has yet to be examined.

Textual analysis uses linguistic, statistical, and machine learning techniques to analyze and extract information from text [26,27]. For instance, scientists analyzed linguistic similarities and differences of biomedical corpora [28–30]. Scientists have provided the community with a number of tools that aide future text mining systems [31–33] as well as advice on how to train and test future text processing systems [34–36]. Here, we use textual analysis to examine the bioRxiv repository, placing a particular emphasis on understanding the extent to which full-text research can address hypotheses derived from the study of metadata alone.

To understand how preprints relate to the traditional publishing ecosystem, we examine the linguistic similarities and differences between preprints and peer-reviewed text and observe how linguistic features change during the peer review and publishing process. We hypothesize that preprints and biomedical text will appear to have similar characteristics, especially when controlling for the differential uptake of preprints across fields. Furthermore, we hypothesize that document embeddings [37,38] provide a versatile way to disentangle linguistic features along with serving as a suitable medium for improving preprint repository functionality. We test this hypothesis by producing a linguistic landscape of bioRxiv preprints, detecting preprints that change substantially during publication, and identifying journals that publish manuscripts that are linguistically similar to a target preprint. We encapsulate our

findings through a web app that projects a user-selected preprint onto this landscape and suggests journals and articles that are linguistically similar. Our work reveals how linguistically similar and dissimilar preprints are to peer-reviewed text, quantifies linguistic changes that occur during the peer review process, and highlights the feasibility of document embeddings concerning preprint repository functionality and peer review's effect on publication time.

## Materials and methods

### Corpora examined

Text analytics is generally comparative in nature, so we selected 3 relevant text corpora for analysis: the bioRxiv corpus, which is the target of the investigation; the PubMed Central Open Access (PMCOA) corpus, which represents the peer-reviewed biomedical literature; and the New York Times Annotated Corpus (NYTAC), which is used a representative of general English text.

### bioRxiv corpus

bioRxiv [7] is a repository for life sciences preprints. We downloaded an XML snapshot of this repository on February 3, 2020, from bioRxiv's Amazon S3 bucket [39]. This snapshot contained the full text and image content of 98,023 preprints. Preprints on bioRxiv are versioned, and in our snapshot, 26,905 out of 98,023 contained more than one version. When preprints had multiple versions, we used the latest one unless otherwise noted. Authors submitting preprints to bioRxiv can select one of 29 different categories and tag the type of article: a new result, confirmatory finding, or contradictory finding. A few preprints in this snapshot were later withdrawn from bioRxiv; when withdrawn, their content is replaced with the reason for withdrawal. We encountered a total of 72 withdrawn preprints within our snapshot. After removal, we were left with 97,951 preprints for our downstream analyses.

### PubMed Central Open Access corpus

PubMed Central (PMC) is a digital archive for the United States National Institute of Health's Library of Medicine (NIH/NLM) that contains full text biomedical and life science articles [40]. Paper availability within PMC is mainly dependent on the journal's participation level [41]. Articles appear in PMC as either accepted author manuscripts (Green Open Access) or via open access publishing at the journal (Gold Open Access [42]). Individual journals have the option to fully participate in submitting articles to PMC, selectively participate sending only a few papers to PMC, only submit papers according to NIH's public access policy [43], or not participate at all; however, individual articles published with the CC BY license may be incorporated. As of September 2019, PMC had 5,725,819 articles available [44]. Out of these 5 million articles, about 3 million were open access (PMCOA) and available for text processing systems [32,45]. PMC also contains a resource that holds author manuscripts that have already passed the peer review process [46]. Since these manuscripts have already been peer reviewed, we excluded them from our analysis as the scope of our work is focused on examining the beginning and end of a preprint's life cycle. We downloaded a snapshot of the PMCOA corpus on January 31, 2020. This snapshot contained many types of articles: literature reviews, book reviews, editorials, case reports, research articles, and more. We used only research articles, which align with the intended role of bioRxiv, and we refer to these articles as the PMCOA corpus.

### The New York Times Annotated Corpus

The NYTAC [47] is a collection of newspaper articles from the New York Times dating from January 1, 1987 to June 19, 2007. This collection contains over 1.8 million articles where 1.5 million of those articles have undergone manual entity tagging by library scientists [47]. We downloaded this collection on August 3, 2020, from the Linguistic Data Consortium (see Software and data availability section) and used the entire collection as a negative control for our corpora comparison analysis.

### Mapping bioRxiv preprints to their published counterparts

We used CrossRef [48] to identify bioRxiv preprints linked to a corresponding published article. We accessed CrossRef on July 7, 2020, and successfully linked 23,271 preprints to their published counterparts. Out of those 23,271 preprint–published pairs, only 17,952 pairs had a published version present within the PMCOA corpus. For our analyses that involved published links, we only focused on this subset of preprints–published pairs.

### Comparing corpora

We compared the bioRxiv, PMCOA, and NYTAC corpora to assess the similarities and differences between them. We used the NYTAC corpus as a negative control to assess the similarity between 2 life sciences repositories compared with nonlife sciences text. All corpora contain multiple words that do not have any meaning (conjunctions, prepositions, etc.) or occur with a high frequency. These words are termed stopwords and are often removed to improve text processing pipelines. Along with stopwords, all corpora contain both words and nonword entities (for instance, numbers or symbols like ±), which we refer to together as tokens to avoid confusion. We calculated the following characteristic metrics for each corpus: the number of documents, the number of sentences, the total number of tokens, the number of stopwords, the average length of a document, the average length of a sentence, the number of negations, the number of coordinating conjunctions, the number of pronouns, and the number of past tense verbs. SpaCy is a lightweight and easy-to-use python package designed to preprocess and filter text [49]. We used spaCy's "en_core_web_sm" model [49] (version 2.2.3) to preprocess all corpora and filter out 326 stopwords using spaCy's default settings.

Following that cleaning process, we calculated the frequency of every token across all corpora. Because many tokens were unique to one set or the other and observed at low frequency, we focused on the union of the top 0.05% (approximately 100) most frequently occurring tokens within each corpus. We generated a contingency table for each token in this union and calculated the odds ratio along with the 95% confidence interval [50]. We measured corpora similarity by calculating the Kullback–Leibler (KL) divergence across all corpora along with token enrichment analysis. KL divergence is a metric that measures the extent to which 2 distributions differ from each other. A low value of KL divergence implicates that 2 distributions are similar and vice versa for high values. The optimal number of tokens used to calculate the KL divergence is unknown, so we calculated this metric using a range of the 100 most frequently occurring tokens between 2 corpora to the 5,000 most frequently occurring tokens.

### Constructing a document representation for life sciences text

We sought to build a language model to quantify linguistic similarities of biomedical preprints and articles. Word2vec is a suite of neural networks designed to model linguistic features of tokens based on their appearance in the text. These models are trained to either predict a token based on its sentence context, called a continuous bag of words (CBOW) model, or

predict the context based on a given token, called a skipgram model [37]. Through these prediction tasks, both networks learn latent linguistic features, which are helpful for downstream tasks, such as identifying similar tokens. We used gensim [51] (version 3.8.1) to train a CBOW [37] model over all the main text within each preprint in the bioRxiv corpus. Determining the best number of dimensions for token embeddings can be a nontrivial task; however, it has been shown that optimal performance is between 100 and 1,000 dimensions [52]. We chose to train the CBOW model using 300 hidden nodes, a batch size of 10,000 tokens, and for 20 epochs. We set a fixed random seed and used gensim's default settings for all other hyperparameters. Once trained, every token present within the CBOW model is associated with a dense vector representing latent features captured by the network. We used these token vectors to generate a document representation for every article within the bioRxiv and PMCOA corpora. We used spaCy to lemmatize each token for each document and then took the average of every lemmatized token present within the CBOW model and the individual document [38]. Any token present within the document but not in the CBOW model is ignored during this calculation process.

## Visualizing and characterizing preprint representations

We sought to visualize the landscape of preprints and determine the extent to which their representation as document vectors corresponded to author-supplied document labels. We used principal component analysis (PCA) [53] to project bioRxiv document vectors into a low-dimensional space. We trained this model using scikit-learn's [54] implementation of a randomized solver [55] with a random seed of 100, an output of 50 principal components (PCs), and default settings for all other hyperparameters. After training the model, every preprint within the bioRxiv corpus receives a score for each generated PC. We sought to uncover concepts captured within generated PCs and used the cosine similarity metric to examine these concepts. This metric takes 2 vectors as input and outputs a score between −1 (most dissimilar) and 1 (most similar). We used this metric to score the similarity between all generated PCs and every token within our CBOW model for our use case. We report the top 100 positive and negative scoring tokens as word clouds. The size of each word corresponds to the magnitude of similarity, and color represents a positive (orange) or negative (blue) association.

## Discovering unannotated preprint–publication relationships

The bioRxiv maintainers have automated procedures to link preprints to peer-reviewed versions, and many journals require authors to update preprints with a link to the published version. However, this automation is primarily based on the exact matching of specific preprint attributes. If authors change the title between a preprint and published version (for instance, [56,57]), then this change will prevent bioRxiv from automatically establishing a link. Furthermore, if the authors do not report the publication to bioRxiv, the preprint and its corresponding published version are treated as distinct entities despite representing the same underlying research. We hypothesize that close proximity in the document embedding space could match preprints with their corresponding published version. If this finding holds, we could use this embedding space to fill in links missed by existing automated processes. We used the subset of paper–preprint pairs annotated in CrossRef as described above to calculate the distribution of available preprint to published distances. We calculated this distribution by taking the Euclidean distance between the preprint's embedding coordinates and the coordinates of its corresponding published version. We also calculated a background distribution, which consisted of the distance between each preprint with an annotated publication and a randomly selected article from the same journal. We compared both distributions to determine if there was a

difference between both groups as a significant difference would indicate that this embedding method can parse preprint–published pairs apart. After comparing the 2 distributions, we calculated distances between preprints without a published version link with PMCOA articles that weren't matched with a corresponding preprint. We filtered any potential links with distances greater than the minimum value of the background distribution as we considered these pairs to be true negatives. Lastly, we binned the remaining pairs based on percentiles from the annotated pairs distribution at the [0,25th percentile), [25th percentile, 50th percentile), [50th percentile, 75th percentile), and [75th percentile, minimum background distance). We randomly sampled 50 articles from each bin and shuffled these 4 sets to produce a list of 200 potential preprint–published pairs with a randomized order. We supplied these pairs to 2 coauthors to manually determine if each link between a preprint and a putative matched version was correct or incorrect. After the curation process, we encountered 8 disagreements between the reviewers. We supplied these pairs to a third scientist, who carefully reviewed each case and made a final decision. Using this curated set, we evaluated the extent to which distance in the embedding space revealed valid but unannotated links between preprints and their published versions.

## Measuring time duration for preprint publication process

Preprints can take varying amounts of time to be published. We sought to measure the time required for preprints to be published in the peer-reviewed literature and compared this time measurement across author-selected preprint categories as well as individual preprints. First, we queried bioRxiv's application programming interface (API) to obtain the date a preprint was posted onto bioRxiv as well as the date a preprint was accepted for publication. We did not include preprint matches found by our paper matching approach (see Discovering unannotated preprint–publication relationships). We measured time elapsed as the difference between the date a preprint was first posted on bioRxiv and its publication date. Along with calculating the time elapsed, we also recorded the number of different preprint versions posted onto bioRxiv.

We used this captured data to apply the Kaplan–Meier estimator [58] via the KaplanMeierFitter function from the lifelines [59] (version 0.25.6) python package to calculate the half-life of preprints across all preprint categories within bioRxiv. We considered survival events as preprints that have yet to be published. We encountered 123 cases where the preprint posting date was subsequent to the publication date, resulting in a negative time difference, as previously reported [60]. We removed these preprints for this analysis as they were incompatible with the rules of the bioRxiv repository.

We measured the textual difference between preprints and their corresponding published version after our half-life calculation by calculating the Euclidean distance for their respective embedding representation. This metric can be difficult to understand within the context of textual differences, so we sought to contextualize the meaning of a distance unit. We first randomly sampled with replacement a pair of preprints from the Bioinformatics topic area as this was well represented within bioRxiv and contains a diverse set of research articles. Next, we calculated the distance between 2 preprints 1,000 times and reported the mean. We repeated the above procedure using every preprint within bioRxiv as a whole. These 2 means serve as normalized benchmarks to compare against as distance units are only meaningful when compared to other distances within the same space. Following our contextualization approach, we performed linear regression to model the relationship between preprint version count with a preprint's time to publication. We also performed linear regression to measure the relationship between document embedding distance and a preprint's time to publication. For this analysis,

we retained preprints with negative time within our linear regression model, and we observed that these preprints had minimal impact on results. We visualize our version count regression model as a violin plot and our document embeddings regression model as a square bin plot.

## Building classifiers to detect linguistically similar journal venues and published articles

Preprints are more likely to be published in journals that publish articles with similar content. We assessed this claim by building classifiers based on document and journal representations. First, we removed all journals that had fewer than 100 papers in the PMC corpus. We held our preprint–published subset (see above section Mapping bioRxiv preprints to their published counterparts) and treated it as a gold standard test set. We used the remainder of the PMCOA corpus for training and initial evaluation for our models.

Training models to identify which journal publishes similar articles is challenging as not all journals are the same. Some journals have a publication rate of at most hundreds of papers per year, while others publish at a rate of at least 10,000 papers per year. Furthermore, some journals focus on publishing articles within a concentrated topic area, while others cover many dispersive topics. Therefore, we designed 2 approaches to account for these characteristics. Our first approach focuses on articles that account for a journal's variation of publication topics. This approach allows for topically similar papers to be retrieved independently of their respective journal. Our second approach is centered on journals to account for varying publication rates. This approach allows more selective or less popular journals to have equal representation to their high publishing counterparts.

Our article-based approach identifies most similar manuscripts to the preprint query, and we evaluated the journals that published these identified manuscripts. We embedded each query article into the space defined by the word2vec model (see above section Constructing a document representation for life sciences text). Once embedded, we selected manuscripts close to the query via Euclidean distance in the embedding space. Once identified, we return articles along with journals that published these identified articles.

We constructed a journal-based approach to accompany the article-based classifier while accounting for the overrepresentation of these high publishing frequency journals. We identified the most similar journals for this approach by constructing a journal representation in the same embedding space. We computed this representation by taking the average embedding of all published papers within a given journal. We then projected a query article into the same space and returned journals closest to the query using the same distance calculation described above.

Both models were constructed using the scikit-learn k-Nearest Neighbors implementation [54] with the number of neighbors set to 10 as this is an appropriate number for our use case. We consider a prediction to be a true positive if the correct journal appears within our reported list of neighbors and evaluate our performance using 10-fold cross-validation on the training set along with test set evaluation.

## Web application for discovering similar preprints and journals

We developed a web application that places any bioRxiv or medRxiv preprint into the overall document landscape and identifies topically similar papers and journals (similar to [61]). Our application attempts to download the full text xml version of any preprint hosted on the bioRxiv or medRxiv server and uses the lxml package (version num) to extract text. If the xml version isn't available our application defaults to downloading the pdf version and uses PyMuPDF [62] to extract text from the pdf. The extracted text is fed into our CBOW model to construct a document embedding representation. We pass this representation onto our journal

and article classifiers to identify journals based on the 10 closest neighbors of individual papers and journal centroids. We implemented this search using the scikit-learn implementation of k-d trees. To run it more cost-effectively in a cloud computing environment with limited available memory, we sharded the k-d trees into 4 trees.

The app provides a visualization of the article's position within our training data to illustrate the local publication landscape. We used SAUCIE [63], an autoencoder designed to cluster single-cell RNA-seq data, to build a two-dimensional embedding space that could be applied to newly generated preprints without retraining, a limitation of other approaches that we explored for visualizing entities expected to lie on a nonlinear manifold. We trained this model on document embeddings of PMC articles that did not contain a matching preprint version. We used the following parameters to train the model: a hidden size of 2, a learning rate of 0.001, lambda_b of 0, lambda_c of 0.001, and lambda_d of 0.001 for 5,000 iterations. When a user requests a new document, we can then project that document onto our generated two-dimensional space, thereby allowing the user to see where their preprint falls along the landscape. We illustrate our recommendations as a shortlist and provide access to our network visualization at our website (https://greenelab.github.io/preprint-similarity-search/).

## Analysis of the Preprints in Motion collection

Our manuscript describes the large-scale analysis of bioRxiv. Concurrent with our work, another set of authors performed a detailed curation and analysis of a subset of bioRxiv [25] that was focused on preprints posted during the initial stages of the COVID-19 pandemic. The curated analysis was designed to examine preprints at a time of increased readership [64] and includes certain preprints posted from January 1, 2020 to April 30, 2020 [25]. We sought to contextualize this subset, which we term "Preprints in Motion" after the title of the preprint [25], within our global picture of the bioRxiv preprint landscape. We extracted all preprints from the set reported in Preprints in Motion [25] and retained any entries in the bioRxiv repository. We manually downloaded the XML version of these preprints and mapped them to their published counterparts as described above. We used PMC's digital object identifier (DOI) converter [65] to map the published article DOIs with their respective PMC IDs. We retained articles that were included in the PMCOA corpus and performed a token analysis as described to compare these preprints with their published versions. As above, we generated document embeddings for every obtained preprint and published article. We projected these preprint embeddings onto our publication landscape to visually observe the dispersion of this subset. We performed a time analysis that paralleled our approach for the full set of preprint–publication pairs to examine relationships between linguistic changes and the time to publication. The "Preprints in Motion" subset includes recent papers, and the longest time to publish in that set was 195 days; however, our bioRxiv snapshot contains both older preprint–published pairs and many with publication times longer than this time point. The optimum comparison would be to consider only preprints posted on the same days as preprints with the "Preprints in Motion" collection. However, based on our results examining publication rate over time, these preprints may not have made it entirely through the publication process. We performed a secondary analysis to control for the time since posting, where we filtered the bioRxiv snapshot to only contain publication pairs with publication time of less than or equal to 195 days.

## Results

### Comparing bioRxiv to other corpora

**bioRxiv metadata statistics.** The preprint landscape is rapidly changing, and the number of bioRxiv preprints in our data download (71,118) was nearly double that of a recent study

**Table 1. Summary statistics for the bioRxiv, PMC, and NYTAC corpora.**

| Metric | bioRxiv | PMC | NYTAC |
|---|---|---|---|
| document count | 71,118 | 1,977,647 | 1,855,658 |
| sentence count | 22,195,739 | 480,489,811 | 72,171,037 |
| token count | 420,969,930 | 8,597,101,167 | 1,218,673,384 |
| stopword count | 158,429,441 | 3,153,077,263 | 559,391,073 |
| avg. document length | 312.10 | 242.96 | 38.89 |
| avg. sentence length | 22.71 | 21.46 | 19.89 |
| negatives | 1,148,382 | 24,928,801 | 7,272,401 |
| coordinating conjunctions | 14,295,736 | 307,082,313 | 38,730,053 |
| coordinating conjunctions% | 3.40% | 3.57% | 3.18% |
| pronouns | 4,604,432 | 74,994,125 | 46,712,553 |
| pronouns% | 1.09% | 0.87% | 3.83% |
| passives | 15,012,441 | 342,407,363 | 19,472,053 |
| passive% | 3.57% | 3.98% | 1.60% |

NYTAC, New York Times Annotated Corpus; PMC, PubMed Central.

that reported on a snapshot with 37,648 preprints [13]. Because the rate of change is rapid, we first analyzed category data and compared our results with previous findings. As in previous reports [13], neuroscience remains the most common category of preprints, followed by bioinformatics (S2 Fig). Microbiology, which was fifth in the most recent report [13], has now surpassed evolutionary biology and genomics to move into third. When authors upload their preprints, they select from 3 result category types: new results, confirmatory results, or contradictory results. We found that nearly all preprints (97.5%) were categorized as new results, consistent with reports on a smaller set [66]. The results taken together suggest that while bioRxiv has experienced dramatic growth, how it is being used appears to have remained consistent in recent years.

**Global analysis reveals similarities and differences between bioRxiv and PMC.** Documents within bioRxiv were slightly longer than those within PMCOA, but both were much longer than those from the control (NYTAC) (Table 1). The average sentence length, the fraction of pronouns, and the use of the passive voice were all more similar between bioRxiv and PMC than they were to NYTAC (Table 1). The KL divergence of term frequency distributions between bioRxiv and PMCOA were low, especially among the top few hundred tokens (Fig 1A). As more tokens were incorporated, the KL divergence started to increase but remained much lower than the biomedical corpora compared against NYTAC. We provide a listing of the top 100 most frequently occurring tokens from all 3 corpora in our supporting information (S4 Table). These findings support our notion that bioRxiv is linguistically similar to the PMCOA repository.

The terms "neurons," "genome," and "genetic," which are common in genomics and neuroscience, were more common in bioRxiv than PMCOA, while others associated with clinical research, such as "clinical," "patients," and "treatment" were more common in PMCOA (Fig 1B, Fig 1C, and S3 Fig). When controlling for the differences in the body of documents to identify textual changes associated with the publication process, we found that tokens such as "et" and "al" were enriched for bioRxiv, while "±" and "−" were enriched for PMCOA (Fig 1D and 1E). When removing special and single-character tokens, data availability and presentation-related terms "file," "supplementary," and "fig" appeared enriched for published articles, and research-related terms "mice," "activity," and "neurons" appeared enriched for bioRxiv

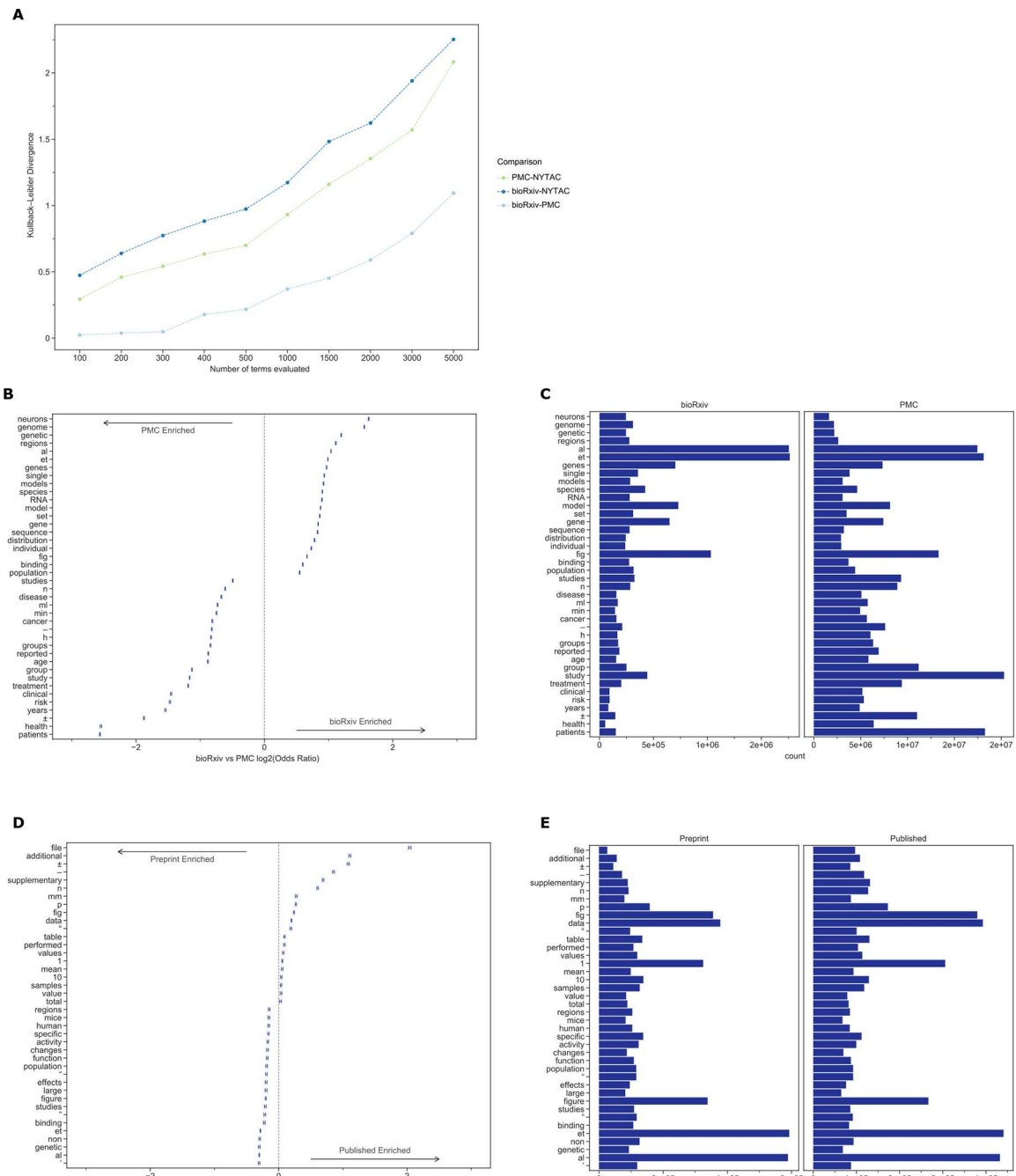

**Fig 1.** (**A**) The KL divergence measures the extent to which the distributions, not specific tokens, differ from each other. The token distribution of bioRxiv and PMC corpora is more similar than these biomedical corpora are to the NYTAC one. (**B**) The significant differences in token frequencies for the corpora appear to be driven by the fields with the highest uptake of bioRxiv, as terms from neuroscience and genomics are relatively more abundant in bioRxiv. We plotted the 95% confidence interval for each reported token. (**C**) Of the tokens that differ between bioRxiv and PMC, the most abundant in bioRxiv are "et" and "al," while the most abundant in PMC is "study." (**D**) The significant differences in token frequencies for preprints and their corresponding published version often appear to be associated with typesetting and supporting information or additional materials. We plotted the 95% confidence interval for each reported token. (**E**) The tokens with the largest absolute differences in abundance appear to be stylistic. Data for the information depicted in this figure are available at https://github.com/greenelab/annorxiver/blob/master/FIGURE_DATA_SOURCE.md#figure-one. KL, Kullback–Leibler; NYTAC, New York Times Annotated Corpus; PMC, PubMed Central.

(S4 Fig). Furthermore, we found that specific changes appeared to be related to journal styles: "figure" was more common in bioRxiv, while "fig" was relatively more common in PMCOA. Other changes appeared to be associated with an increasing reference to content external to the manuscript itself: the tokens "supplementary," "additional," and "file" were all more common in PMCOA than bioRxiv, suggesting that journals are not simply replacing one token with another but that there are more mentions of such content after peer review.

These results suggest that the text structure within preprints on bioRxiv is similar to published articles within PMCOA. The differences in uptake across fields are supported by the authors' categorization of their articles and the text within the articles themselves. At the level of individual manuscripts, the most change terms appear to be associated with typesetting, journal style, and an increasing reliance on additional materials after peer review.

Following our analysis of tokens, we examined the PCs of document embeddings derived from bioRxiv. We found that the top PCs separated methodological approaches and research fields. Preprints from certain topic areas that spanned approaches from informatics-related to cell biology could be distinguished using these PCs (see S1 Text).

## Document embedding similarities reveal unannotated preprint–publication pairs

Distances between preprints and their corresponding published versions were nearly always lower than preprints paired with a random article published in the same journal (Fig 2A). This suggested that embedding distances may predict the published form of preprints. We directly tested this by selecting low-distance but unannotated preprint–publication pairs and curating the extent to which they represented matching documents. Approximately 98% of our 200 pairs with an embedding distance in the 0 to 25th and 25th to 50th percentile bins were successfully matched with their published counterpart (Fig 2B). These 2 bins contained 1,542 preprint–article pairs, suggesting that many preprints may have been published but not previously connected with their published versions. There is a particular enrichment for preprints published but unlinked within the 2017 to 2018 interval (Fig 2C). We expected a higher proportion of such preprints before 2019 (many of which may not have been published yet); however, observing relatively few missed annotations before 2017 was against our expectations. There are several possible explanations for this increasing fraction of missed annotations. As the number of preprints posted on bioRxiv grows, it may be harder for bioRxiv to establish a link between preprints and their published counterparts simply due to the scale of the challenge. It is possible that the set of authors participating in the preprint ecosystem is changing and that new participants may be less likely to report missed publications to bioRxiv. Finally, as familiarity with preprinting grows, it is possible that authors are posting preprints earlier in the process and that metadata fields that bioRxiv uses to establish a link may be less stable.

## Preprints with more versions or more text changes relative to their published counterpart took longer to publish

The process of peer review includes several steps, which take variable amounts of time [67], and we sought to measure if there is a difference in publication time between author-selected categories of preprints (Fig 3A). Of the most abundant preprint categories, microbiology was the fastest to publish (140 days, (137, 145 days) [95% CI]), and genomics was the slowest (190 days, (185, 195 days) [95% CI]) (Fig 3A). We did observe category-specific differences; however, these differences were generally modest, suggesting that the peer review process did not differ dramatically between preprint categories. One exception was the Scientific

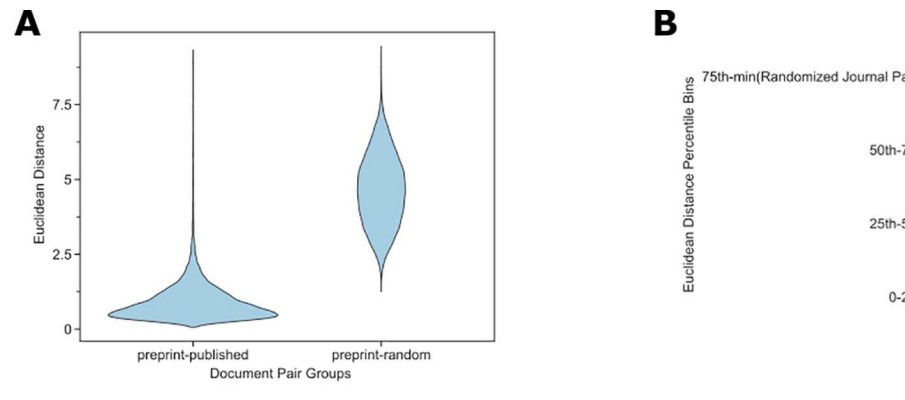
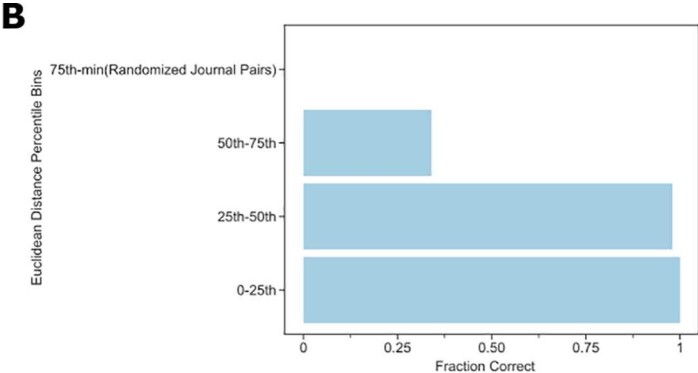

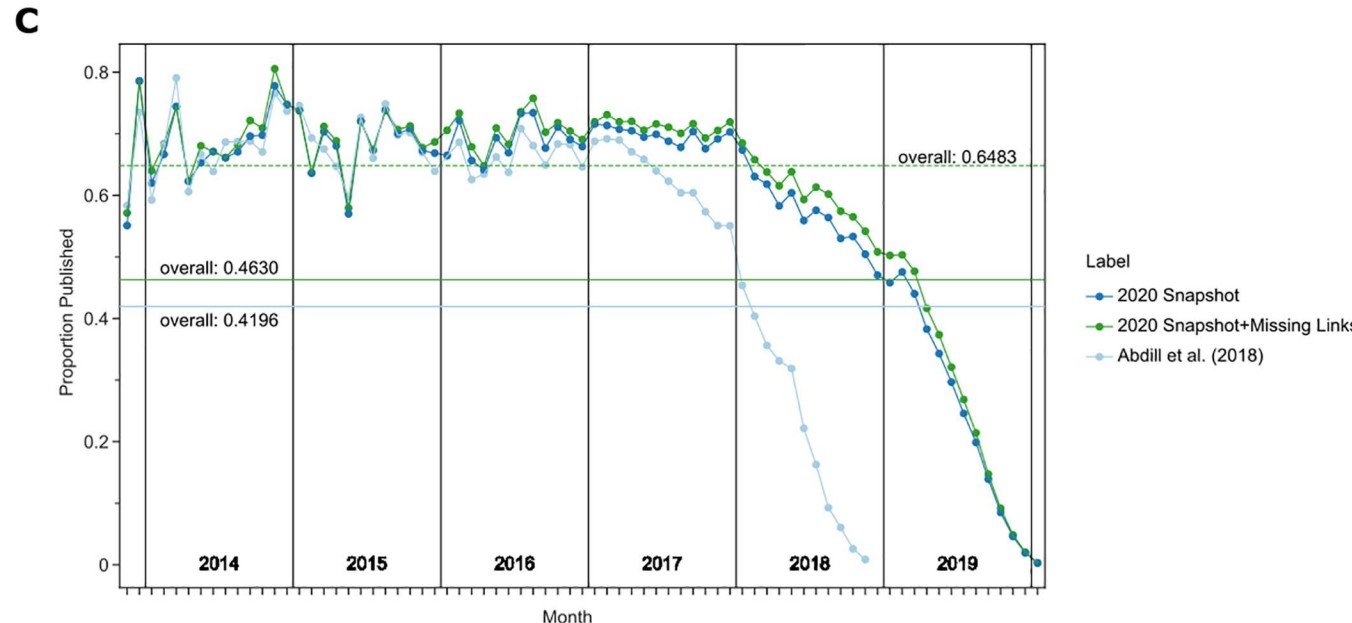

**Fig 2.** (**A**) Preprints are closer in document embedding space to their corresponding peer-reviewed publication than they are to random papers published in the same journal. (**B**) Potential preprint–publication pairs that are unannotated but within the 50th percentile of all preprint–publication pairs in the document embedding space are likely to represent true preprint–publication pairs. We depict the fraction of true positives over the total number of pairs in each bin. Accuracy is derived from the curation of a randomized list of 200 potential pairs (50 per quantile) performed in duplicate with a third rater used in the case of disagreement. (**C**) Most preprints are eventually published. We show the publication rate of preprints since bioRxiv first started. The x-axis represents months since bioRxiv started, and the y-axis represents the proportion of preprints published given the month they were posted. The light blue line represents the publication rate previously estimated by Abdill and colleagues [13]. The dark blue line represents the updated publication rate using only CrossRef-derived annotations, while the dark green line includes annotations derived from our embedding space approach. The horizontal lines represent the overall proportion of preprints published as of the time of the annotated snapshot. The dashed horizontal line represents the overall proportion published preprints for preprints posted before 2019. Data for the information depicted in this figure are available at https://github.com/greenelab/annorxiver/blob/master/FIGURE_DATA_SOURCE.md#figure-two.

Communication and Education category, which took substantially longer to be peer reviewed and published (373 days, (373, 398 days) [95% CI]). This hints that there may be differences in the publication or peer review process or culture that apply to preprints in this category.

Examining peer review's effect on individual preprints, we found a positive correlation between preprints with multiple versions and the time elapsed until publication (Fig 3B). Every additional preprint version was associated with an increase of 51 days before a preprint was published. This time duration seems broadly compatible with the amount of time it would

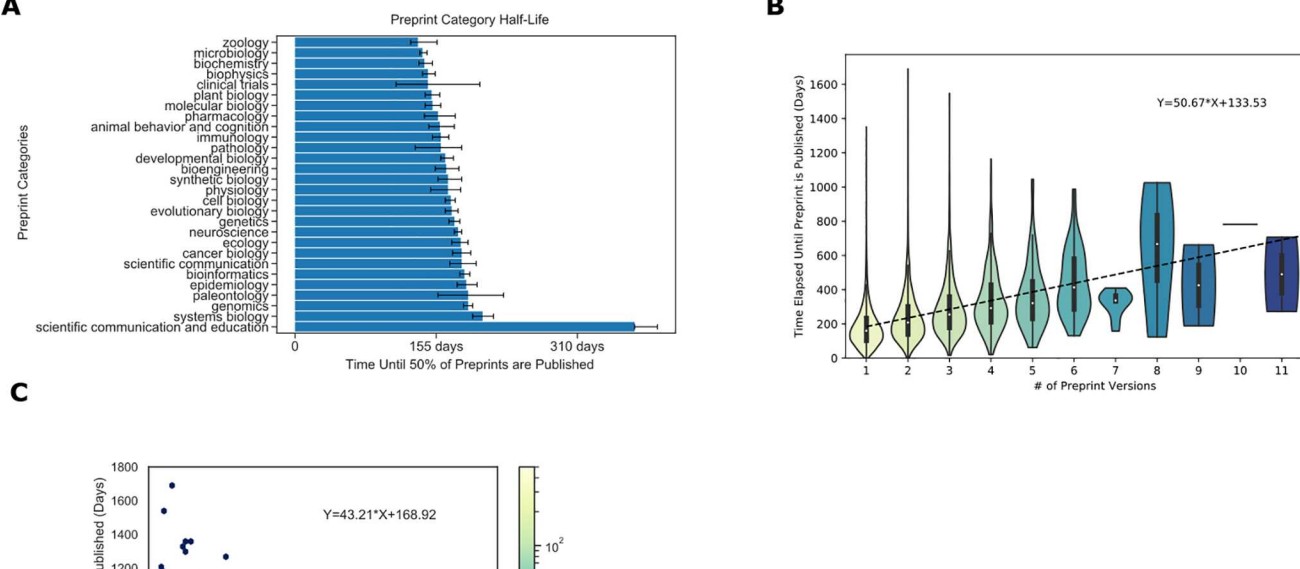

**Fig 3. (A)** Author-selected categories were associated with modest differences in the median time to publish. Author-selected preprint categories are shown on the y-axis, while the x-axis shows the median time-to-publish for each category. Error bars represent 95% confidence intervals for each median measurement. **(B)** Preprints with more versions were associated with a longer time to publish. The x-axis shows the number of versions of a preprint posted on bioRxiv. The y-axis indicates the number of days that elapsed between the first version of a preprint posted on bioRxiv and the date at which the peer-reviewed publication appeared. The density of observations is depicted in the violin plot with an embedded boxplot. **(C)** Preprints with more substantial text changes took longer to be published. The x-axis shows the Euclidean distance between document representations of the first version of a preprint and its peer-reviewed form. The y-axis shows the number of days elapsed between the first version of a preprint posted on bioRxiv and when a preprint is published. The color bar on the right represents the density of each hexbin in this plot, where more dense regions are shown in a brighter color. Data for the information depicted in this figure are available at https://github.com/greenelab/annorxiver/blob/master/FIGURE_DATA_SOURCE.md#figure-three.

take to receive reviews and revise a manuscript, suggesting that many authors may be updating their preprints in response to peer reviews or other external feedback. The embedding space allows us to compare preprint and published documents to determine if the level of change that documents undergo relates to the time it takes them to be published. Distances in this space are arbitrary and must be compared to reference distances. We found that the average distance of 2 randomly selected papers from the bioinformatics category was 4.470, while the average distance of 2 randomly selected papers from bioRxiv was 5.343. Preprints with large embedding space distances from their corresponding peer-reviewed publication took longer to publish (Fig 3C): Each additional unit of distance corresponded to roughly 43 additional days.

Overall, our findings support a model where preprints are reviewed multiple times or require more extensive revisions take longer to publish.

## Preprints with similar document embeddings share publication venues

We developed an online application that returns a listing of published papers and journals closest to a query preprint in document embedding space. This application uses 2 k-nearest

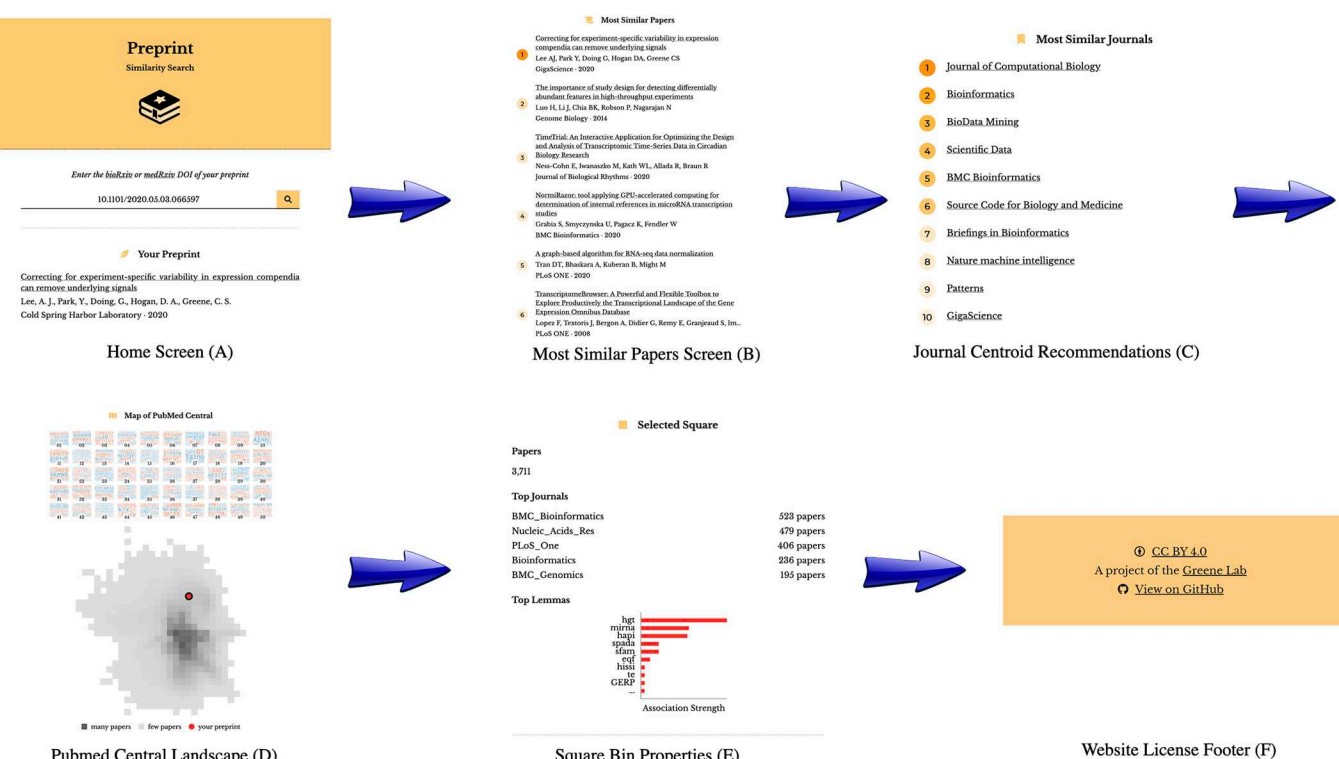

**Fig 4. The preprint-similarity-search app workflow allows users to examine where an individual preprint falls in the overall document landscape.** (A) Starting with the home screen, users can paste in a bioRxiv or medRxiv DOI, which sends a request to bioRxiv or medRxiv. Next, the app preprocesses the requested preprint and returns a listing of (B) the top 10 most similar papers and (C) the 10 closest journals. (D) The app also displays the location of the query preprint in PMC. (E) Users can select a square within the landscape to examine statistics associated with the square, including the top journals by article count in that square and the odds ratio of tokens. DOI, digital object identifier; PMC, PubMed Central.

neighbor classifiers that achieved better performance than our baseline model (S5 Fig) to identify these entities. Users supply our app with DOIs from bioRxiv or medRxiv, and the corresponding preprint is downloaded from the repository. Next, the preprint's PDF is converted to text, and this text is used to construct a document embedding representation. This representation is supplied to our classifiers to generate a listing of the 10 papers and journals with the most similar representations in the embedding space (Fig 4A–4C). Furthermore, the user-requested preprint's location in this embedding space is then displayed on our interactive map, and users can select regions to identify the terms most associated with those regions (Fig 4D and 4E). Users can also explore the terms associated with the top 50 PCs derived from the document embeddings, and those PCs vary across the document landscape. You can access this application using the following url: https://greenelab.github.io/preprint-similarity-search/

## Contextualizing the Preprints in Motion collection

The Preprints in Motion collection included a set of preprints posted during the first 4 months of 2020. We examined the extent to which preprints in this set were representative of the patterns that we identified from our analysis on all of bioRxiv. As with all of bioRxiv, typesetting tokens changed between preprints and their paired publications. Our token-level analysis identified certain patterns consistent with our findings across bioRxiv (Fig 5A and 5B). However, in this set, we also observe changes likely associated with the fast-moving nature of COVID-19 research: The token "2019-ncov" became less frequently represented, while "sars"

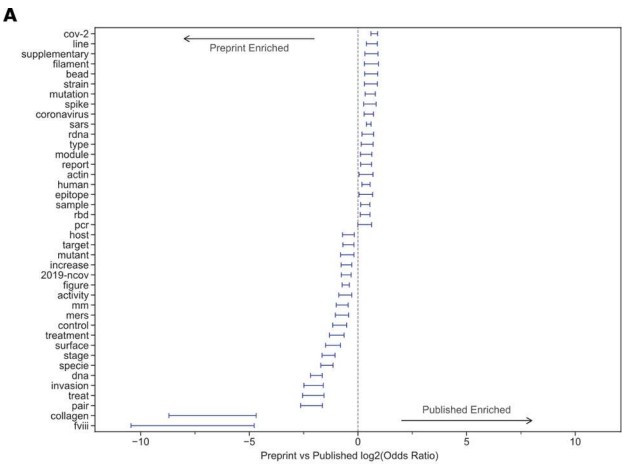

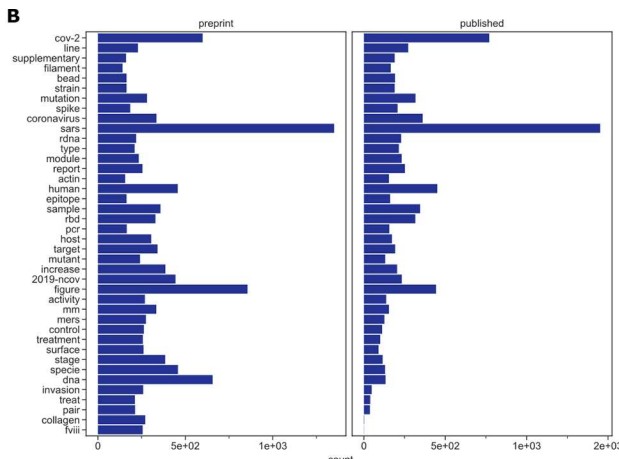

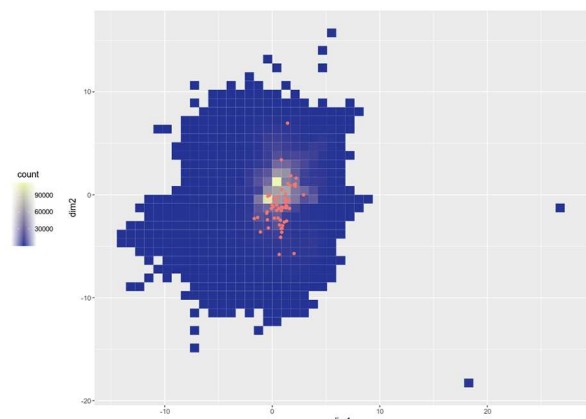

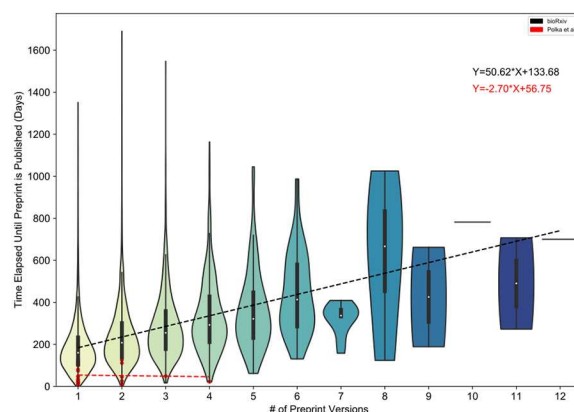

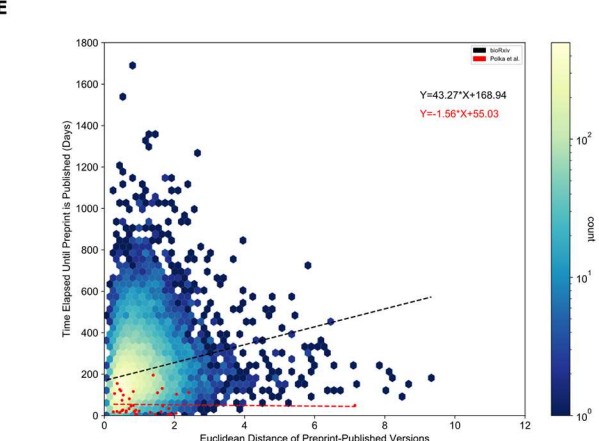

**Fig 5. The Preprints in Motion collection results are similar to all preprint results, except that their time to publication was independent of the number of preprint versions and amount of linguistic change.** (A) Tokens that differed included those associated with typesetting and those related to the nomenclature of the virus that causes COVID-19. Error bars show 95% confidence intervals for each token. (B) Of the tokens that differ between Preprints in Motion and their published counterparts, the most abundant were associated with the nomenclature of the virus. (C) The Preprints in Motion collection fall

across the landscape of PMCOA with respect to linguistic properties. This square bin plot depicts the binning of all published papers within the PMCOA corpus. High-density regions are depicted in yellow, while low-density regions are in dark blue. Red dots represent the Preprints in Motion collection. (D) The Preprints in Motion collection were published faster than other bioRxiv preprints, and the number of versions was not associated with an increase in time to publication. The x-axis shows the number of versions of a preprint posted on bioRxiv. The y-axis indicates the number of days that elapsed between the first version of a preprint posted on bioRxiv and the date at which the peer-reviewed publication appeared. The density of observations is depicted in the violin plot with an embedded boxplot. The red dots and red regression line represent Preprints in Motion. (E) The Preprints in Motion collection were published faster than other bioRxiv preprints, and no dependence between the amount of linguistic change and time to publish was observed. The x-axis shows the Euclidean distance between document representations of the first version of a preprint and its peer-reviewed form. The y-axis shows the number of days elapsed between the first version of a preprint posted on bioRxiv and when a preprint is published. The color bar on the right represents the density of each hexbin in this plot, where more dense regions are shown in a brighter color. The red dots and red regression line represent Preprints in Motion. Data for the information depicted in this figure are available at https://github.com/greenelab/annorxiver/blob/master/FIGURE_DATA_SOURCE.md#figure-five. PMCOA, Pubmed Central's Open Access.

and "cov-2" became more represented, likely due to a shift in nomenclature from "2019-nCoV" to "SARS-CoV-2". The Preprints in Motion were not strongly colocalized in the linguistic landscape, suggesting that the collection covers a diverse set of research approaches (Fig 5C). Preprints in this collection were published faster than the broader set of bioRxiv preprints (Fig 5D and 5E). We see the same trend when filtering the broader bioRxiv set to only contain preprints published within the same time frame as this collection (S6A and S6B Fig). The relationship between time to publication and the number of versions (Fig 5D and S6A Fig) and the relationship between time to publication and the amount of linguistic change (Fig 5E and S6B Fig) were both lost in the Preprints in Motion set. Our findings suggest that Preprints in Motion changed during publication in ways aligned with changes in the full preprint set but that peer review was accelerated in ways that broke the time dependencies observed with the full bioRxiv set.

## Discussion and conclusions

bioRxiv is a constantly growing repository that contains life science preprints. Over 77% of bioRxiv preprints with a corresponding publication in our snapshot were successfully detected within PMCOA corpus. This suggests that most work from groups participating in the preprint ecosystem is now available in final form for literature mining and other applications. Most research on bioRxiv preprints has examined their metadata; we examine the text content as well. Throughout this work, we sought to analyze the language within these preprints and understand how it changes in response to peer review.

Our global corpora analysis found that writing within bioRxiv is consistent with the biomedical literature in the PMCOA repository, suggesting that bioRxiv is linguistically similar to PMCOA. Token-level analyses between bioRxiv and PMCOA suggested that research fields drive significant differences; for instance, more patient-related research is prevalent in PMCOA than bioRxiv. This observation is expected as preprints focused on medicine are supported by the complementary medRxiv repository [8]. Token-level analyses for preprints and their corresponding published version suggest that peer review may focus on data availability and incorporating extra sections for published papers; however, future studies are needed to ascertain individual token level changes as preprints venture through the publication process. One future avenue of research could examine the differences between only preprints and accepted author manuscripts within PMC to identify changes prior to journal publication.

Document embeddings are a versatile way to examine language contained within preprints, understand peer review's effect on preprints, and provide extra functionality for preprint repositories. Our approach to generate document embeddings was focused on interpretability instead of predictive performance; however, using more advanced strategies to generate document vectors such as Doc2Vec [38] or BERT [68] should increase predictive performance.

Examining linguistic variance within document embeddings of life science preprints revealed that the largest source of variability was informatics. This observation bisects the majority of life science research categories that have integrated preprints within their publication workflow. This embedding space could also be used to quantify sentiment trends or other linguistic features. Furthermore, methodologies for uncovering latent scientific knowledge [69] may be applicable in this embedding space.

Preprints are typically linked with their published articles via bioRxiv manually establishing links or authors self-reporting that their preprint has been published; however, gaps can occur as preprints change their appearance through multiple versions or authors do not notify bioRxiv. Our work suggests that document embeddings can help fill in missing links within bioRxiv.

Furthermore, our analysis reveals that the publication rate for preprints is higher than previously estimated, even though our analysis can only account for published open access papers. Our results raise the lower bound of the total preprint publication fraction; however, the true fraction is necessarily higher. Future work, especially that which aims to assess the fraction of preprints that are eventually published, should account for the possibility of missed annotations.

Preprints take a variable amount of time to become published, and we examined factors that influence a preprint's time to publication. Our half-life analysis on preprint categories revealed that preprints in most bioRxiv categories take similar amounts of time to be published. An apparent exception is the scientific communication and education category, which contained preprints that took much longer to publish. Regarding individual preprints, each new version adds several weeks to a preprints time to publication, which is roughly aligned with authors making changes after a round of peer review; furthermore, preprints that undergo substantial changes take longer to publish. Overall, these results illustrate that bioRxiv is a practical resource for obtaining insight into the peer review process.

Lastly, we found that document embeddings were associated with the eventual journal at which the work was published. We trained 2 machine learning models to identify which journals publish linguistically similar papers toward a query preprint. Our models achieved a considerably higher fold change over the baseline model, so we constructed a web application that makes our models available to the public and returns a list of the papers and journals that are linguistically similar to a bioRxiv or medRxiv preprint.

## Supporting information

**S1 Text. Document embeddings derived from bioRxiv reveal fields and subfields.**
(DOCX)

**S1 Data. Listing of published preprints and their corresponding publication times.**
(XLSX)

**S1 Table. PC1 divided the author-selected category of systems biology preprints along an axis from computational to molecular approaches.**
(DOCX)

**S2 Table. Top and bottom 5 cosine similarity scores between tokens and the PC1 axis.**
(TSV)

**S3 Table. Top and bottom 5 cosine similarity scores between tokens and the PC2 axis.**
(TSV)

**S4 Table. The top 100 frequently occurring tokens across our three corpora.**
(TSV)

**S1 Fig.** (**A**) PCA of bioRxiv word2vec embeddings groups documents based on author-selected categories. We visualized documents from key categories on a scatterplot for the first 2 PCs. The first PC separated cell biology from informatics-related fields, and the second PC separated bioinformatics from neuroscience fields. (**B**) A word cloud visualization of PC1. Each word cloud depicts the cosine similarity score between tokens and the first PC. Tokens in orange were most similar to the PC's positive direction, while tokens in blue were most similar to the PC's negative direction. The size of each token indicates the magnitude of the similarity. (**C**) A word cloud visualization of PC2, which separated bioinformatics from neuroscience. Similar to the first PC, tokens in orange were most similar to the PC's positive direction, while tokens in blue were most similar to the PC's negative direction. The size of each token indicates the magnitude of the similarity. (**D**) Examining PC1 values for each article by category created a continuum from informatics-related fields on the top through cell biology on the bottom. Specific article categories (neuroscience and genetics) were spread throughout PC1 values. (**E**) Examining PC2 values for each article by category revealed fields like genomics, bioinformatics, and genetics on the top and neuroscience and behavior on the bottom. PC, principal component; PCA, principal component analysis.
(TIFF)

**S2 Fig. Neuroscience and bioinformatics are the 2 most common author-selected topics for bioRxiv preprints.**
(TIFF)

**S3 Fig.** (**A**) The significant differences in token frequencies for the corpora appear to be driven by the fields with the highest uptake of bioRxiv, as terms from neuroscience and genomics are relatively more abundant in bioRxiv. We plotted the 95% confidence interval for each reported token. (**B**) Of the tokens that differ between bioRxiv and PMC, the most abundant in bioRxiv are "gene," "genes," and "model," while the most abundant in PMC is "study." PMC, PubMed Central.
(TIFF)

**S4 Fig.** (**A**) The significant differences in token frequencies for preprints and their corresponding published version often appear to be associated with data availability and supporting information or additional materials. We plotted the 95% confidence interval for each reported token. (**B**) The tokens with the largest absolute differences in abundance appear related to scientific figures and data availability.
(TIFF)

**S5 Fig. Both classifiers outperform the randomized baseline when predicting a paper's journal endpoint.** This bargraph shows each model's accuracy in respect to predicting the training and test set.
(TIFF)

**S6 Fig.** (**A**) The Preprints in Motion were published faster than other bioRxiv preprints, and the number of versions was not associated with an increase in time to publication. The x-axis shows the number of versions of a preprint posted on bioRxiv. The y-axis indicates the number of days that elapsed between the first version of a preprint posted on bioRxiv and the date at which the peer-reviewed publication appeared. The density of observations is depicted in the violin plot with an embedded boxplot. The red dots and red regression line represent Preprints in Motion. (**B**) The Preprints in Motion collection were published faster than other bioRxiv

preprints, and no dependence between the amount of linguistic change and time to publish was observed. The x-axis shows the Euclidean distance between document representations of the first version of a preprint and its peer-reviewed form. The y-axis shows the number of days elapsed between the first version of a preprint posted on bioRxiv and when a preprint is published. The color bar on the right represents the density of each hexbin in this plot, where more dense regions are shown in a brighter color. The red dots and red regression line represent Preprints in Motion.
(TIFF)

## Acknowledgments

The authors would like to thank Ariel Hippen Anderson for evaluating potential missing preprint to published version links. We also would like to thank Richard Sever and the bioRxiv team for their assistance with access to and support with questions about preprint full text downloaded from bioRxiv.

The opinions expressed here do not reflect the official policy or positions of Elsevier Inc.

## Author Contributions

**Conceptualization:** David N. Nicholson, Marvin Thielk, Lawrence E. Hunter, Casey S. Greene.

**Data curation:** David N. Nicholson, Marvin Thielk, Casey S. Greene.

**Formal analysis:** David N. Nicholson, Marvin Thielk.

**Funding acquisition:** David N. Nicholson, Casey S. Greene.

**Investigation:** David N. Nicholson, Dongbo Hu.

**Methodology:** David N. Nicholson, Marvin Thielk.

**Project administration:** Casey S. Greene.

**Resources:** Vincent Rubinetti, Dongbo Hu.

**Software:** David N. Nicholson, Vincent Rubinetti, Dongbo Hu.

**Supervision:** Casey S. Greene.

**Validation:** David N. Nicholson.

**Visualization:** David N. Nicholson, Vincent Rubinetti, Dongbo Hu.

**Writing – original draft:** David N. Nicholson.

**Writing – review & editing:** David N. Nicholson, Vincent Rubinetti, Dongbo Hu, Marvin Thielk, Lawrence E. Hunter, Casey S. Greene.

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
