## [Editor Report · Decision Letter 0]

21 May 2021

Dear Dr Greene, 

Thank you for submitting your manuscript entitled "Linguistic analysis of the bioRxiv preprint landscape" for consideration as a Meta-Research Article by PLOS Biology.

Your manuscript has now been evaluated by the PLOS Biology editorial staff, as well as by an academic editor with relevant expertise, and I'm writing to let you know that we would like to send your submission out for external peer review.

Please re-submit your manuscript within two working days, i.e. by May 25 2021 11:59PM.

Kind regards,

Roli Roberts

Roland Roberts

Senior Editor

PLOS Biology

rroberts@plos.org

---

## [Decision Letter · Decision Letter 1]

16 Jul 2021

Dear Dr Greene,

Thank you very much for submitting your manuscript "Linguistic analysis of the bioRxiv preprint landscape" for consideration as a Meta-Research Article at PLOS Biology. Your manuscript has been evaluated by the PLOS Biology editors, an Academic Editor with relevant expertise, and by three independent reviewers.

You'll see that all three reviewers are positive about your study, and while there are a few additional analyses requested by reviewers #2 and #3, the remaining requests are largely textual and/or presentational.

In light of the reviews (below), we are pleased to offer you the opportunity to address the comments from the reviewers in a revised version that we anticipate should not take you very long. We will then assess your revised manuscript and your response to the reviewers' comments and we may consult the reviewers again.

We expect to receive your revised manuscript within 2 months.

**IMPORTANT - SUBMITTING YOUR REVISION**

*Resubmission Checklist*

*Published Peer Review*

*PLOS Data Policy*

*Blot and Gel Data Policy*

Sincerely,

Roli Roberts

Roland Roberts

Senior Editor

PLOS Biology

rroberts@plos.org

REVIEWERS' COMMENTS:

Reviewer #1:

[identifies himself as Ross Mounce]

This manuscript 'Linguistic Analysis of the bioRxiv Preprint Landscape' presents an analysis of bioRxiv fulltexts and metadata, relative to journal-published versions of the same preprints (n= 17,952 pairs), and the New York Times Annotated Corpus.

It's an interesting manuscript worthy of publication in PLOS Biology after a few relatively minor revisions.

I have left many comments directly on the manuscript via the dedicated manuscript website, using public Hypothes.is annotations: https://greenelab.github.io/annorxiver_manuscript/

I incorporate some but not all of these comments into this formal review supplied to the journal (PLOS Biology) who invited me to review this manuscript.

Unsurprisingly, biorxiv preprints and journal-published versions of biorxiv preprints are found to be linguistically different to the New York Times Annotated Corpus e.g. in average document length and to a lesser degree in average sentence length, and % in passive voice.

Luckily there are plenty more actually interesting results reported in this manuscript, not least that of 23,271 preprint-published pairs, 17,952 of those pairs (>77%) had a published version present within the PMCOA corpus. I don't think the authors quite realise the significance of this result. 77% is a very very high rate of open access. It could do with being discussed more within the manuscript e.g. relative to the overall (lower) rate of open access of _all_ biomedical and life science research articles. What does this signify about preprint authors / 'preprinters'? I can think of a couple of hypotheses: a) preprinters are perhaps more likely to have grant-funded research subject to an open access policy

b) perhaps preprinters are more publishing 'savvy' and want to achieve more impact/citations and thus strive harder to ensure that the eventual journal-published version of their work is open access (reflected in being in PMCOA).

If it were my choice I would cut the entire subsection 'Document embeddings derived from bioRxiv reveal fields and subfields'. It is already known that document embeddings can reveal fields and subfields. Being 'preprints' or 'biorxiv preprints' rather than say published journal articles won't change that. I found this section very uninteresting and extremely un-novel. It is descriptive and accurate, but in the context of an already long manuscript, I feel it is unnecessary.

Aside from the manuscript, I have some brief comments on the actual web application.

I tried some palaeontology preprints (it's a field i'm very familiar with). The results were rather mixed.

e.g. for https://greenelab.github.io/preprint-similarity-search/?doi=10.1101/2020.12.10.406678 ("The first dinosaur egg remains a mystery"), the most similar paper recommendations were excellent. However, the most similar journals suggested were surprisingly poor - many of these could obviously at a glance never publish this preprint (dinosaurs are not plants!) e.g. American Journal of Botany, World Archaeology, Journal of Phycology, The Holocene, Botanical Journal of the Linnean Society. Linnean Society of London

But I realise that PMCOA isn't exactly great training date for interpreting palaeontology articles -- fringe content from PMC's perspective(?)

Specific comments:

1.) https://hypothes.is/a/1ODs5NLbEeuhnEPjCpUFpA

2.) https://hypothes.is/a/djNirNLcEeu2YxNM5WBxvA

3.) https://hypothes.is/a/bmdkpNLeEeuZ0k8oLqyeWQ

4.) https://hypothes.is/a/-v_PbtLeEeu8Rw8afgzT5g

5.) https://hypothes.is/a/vekhYNLfEeuxGm9i6MkLqw

it's a real pity you chose not to compare preprints to author manuscripts. As your results demonstrated, lots of the word changes were just journal-style related e.g. "figure" -> "fig." . An analysis of just preprints matched to author manuscripts would get more closely and cleanly to what the textual difference between pre-peer-review and post-peer-review (without minor stylistic changes).

6.) https://hypothes.is/a/YkqPKNLgEeucPGc1_W-jzA

7.) https://hypothes.is/a/Bsv68tLkEeugHEfCZMUGag

8.) https://hypothes.is/a/aWmt3tLlEeuh3M-Ku8ThvQ

9.) https://hypothes.is/a/dkIurtLmEeuqkU-qTN4dtg

10.) https://hypothes.is/a/-2AU8tLmEeu6swvY4jN5Mw

11.) https://hypothes.is/a/dhGbgNLnEeuYkkfUNqKwgA

12.) https://hypothes.is/a/DzrzGtLtEeuJlcdOlUfUBA

13.) https://hypothes.is/a/eJ7-DtLpEeuh0A8OL8WAWw

14.) https://hypothes.is/a/-ixA_NLqEeuv6YviR-lI0A

15.) https://hypothes.is/a/V2wj3NLqEeuZ9idQ44yS4Q

16.) https://hypothes.is/a/N8cKltLrEeuZ-iPc5lrP5g

17.) https://hypothes.is/a/z4r5ztLgEeupmPfOPmlVbw

is using a proprietary data set that charges for access congruent with the PLOS data availability policy?

See: "Please note, if data have been obtained from a third-party source, we require that other researchers would be able to access the data set in the same manner as the authors" https://journals.plos.org/plosone/s/data-availability despite that URL indicating just PLOS ONE, the policy applies to all PLOS journals, unless otherwise noted.

Reviewer #2:

Overall, I enjoyed this manuscript for offering a way to quantify the transition of preprints to manuscripts within the biological sciences. Further, the authors develop an approach that could also more generally be useful for classifying biomedical literature, and they even provide as an example a web-based program to find potential publication avenues. 

The methodological approach of the authors is quite unexpected. While I do not see a fundamental flaw in their approach, I would anticipate it to be biased toward the most frequent phrases. When performing computational research there is a risk to pursue analyses through well-intended "improvements" or "customizations" whenever the approach does not seem to yield the expected outcome. As some people could be tempted to interpret parts of the analysis of the authors as warning flags for above having happened, I would recommend adding some additional control analyses and explicit statements about their chosen rationales.

Particularly, I would be very curious about the discussion, or main text commenting on why the authors created a custom scheme of classifying documents and their similarity based on vectors of words instead of using existing approaches that provide vectors of documents - including doc2vec that is included in the software package that the authors used for word2vec. Do the results change according to the approach?

Further, word2vec often seems to work even better when first trained on a larger corpus before then being applied or transferred to more specialized corpora. Personally, I also made this experience when following an example tutorial provided by the creators of the package that the authors used - which too suggests starting with existing pre-trained models. While the more restricted training done by the authors might have reduced the sensitivity of their approach (… which would likely only strengthen their claims), I would be curious whether there was an additional rationale for avoiding the former strategy that might be missed by readers (e.g.: different meanings such as "abstract" that has different meanings for scientists and non-scientists?). 

Likewise, I'm wondering why the authors used a Euclidean distance for word embeddings instead of a Cosine similarity (which if I recall correctly would also be default in the similarity module of the package which the authors used). Cosine similarity should also allow the authors to make statements about the similarity of words without imposing assumptions on similar text lengths or usage frequencies. 

Similarly, I was wondering how the "journal-based" approach, which the authors mention briefly against the influence of high publication frequency journals, was implemented. Further, if it could have been avoided by avoiding the Euclidean space.

The mapping of similarity seems to be based on individual pairs of text and as such it would seem vulnerable of shifting distributions (e.g.: if published articles were somewhat different from preprints, as implied in Figure 1A). I would suspect that the authors would be able to improve their performance even further by doing global matching between many pairs (… again see their adherence to a weaker approach as something that ultimately strengthens their findings). Again, a comment on the rationale of their chosen approach could convey additional non-evident considerations. 

I love the web application!

No statistics are given for the enrichments in Figure 1B-E.

I would welcome a supplemental analysis, that removes single letters and special characters from the analysis of Figure 1B-E as they might change the baseline. 

The word cloud of Figure 2B, C is somewhat nice as it shows the main words. However, this information could also be conveyed in the text. Would personally favor to quantitatively see loadings of first few principal components for different terms.

The definition of "True matches" could be more explicit in within the main text as the preceding figure 3A could for some people set up a different anticipation.

The association given in Figure 4A seems to mainly stem from a few papers with large distances. Would an association be present when using the rank-based Spearman correlation instead of a linear regression? Would, for visualization, a logarithmic relationship describe the data better than a linear one?

I believe that the analysis of Figure 4 B is quite clever as it would seem to address the thinkable concern of preprints with no delay and changes mainly stemming from those manuscripts that were already essentially accepted by manuscripts at the time of posting.

The analysis remarks that for the "Preprints in Motion Collection" the relationship between textual distance and time to publication disappears, and supports this through Figure 6E. However, the background trend in figure 6E includes publications that have been published at a time that exceeds a year. Hence a more faithful comparison would be to censor the background data by a distribution of durations that would correspond to the distribution of durations that would be possible for the "Preprint in Motions Collection" (taking distribution corresponding to interval between their dates on bioRxiv and the time at which authors assessed whether manuscripts were published).

Other: 

Labels within figures could often be increased in size to improve readability. 

The methods section briefly comments on some ambiguous cases for the matching. Would these cases be the result of modifications that defy a 1:1 mapping, e.g.: multiple stories getting fused, or one story getting split?

The results of Figure 2A could possibly be strengthened by avoiding Principal Components and replacing them by UMAP projects to account for non-linearity.

Although peripheral to the current manuscript, their approach and data would also seem capable to providing an update-able map of the biomedical sciences, by applying their approach of Figure 2 to the PMC corpus data which the authors access too. Such a map could be interesting for those trying to obtain an overview about biology. In case that the authors do not hold plans to publish this elsewhere, and in case that it would be less than a day of work, I would recommend adding such a map to the supplement or as a web service. 

Are the few publications in Figure 2A, which lie outside of the space that is generally occupied by their respective article categories, somewhat different when doing a superficial manual inspection (e.g.: misclassified by authors, or interdisciplinary research)

Adding a few words to "examining the top five and bottom five preprints" could avoid misunderstanding (e.g.: while I suspect that it is the position in Figure 2A, I was first thinking about the most/least successful/downloaded…)

The vector representation of words and documents should allow the authors to quantify shifts that appear between preprints and published manuscripts. Though not necessary from my perspective, many interesting analyses could be done in vector space (e.g.: does language get more positive, or start to refer to more established concepts…?). Maybe there is something small that could be done. Alternatively, the discussion could possibly be extended to demonstrate the implications of vector space, and thus their own work, for future research into preprints and peer review.

Along above, the discussion could be extended toward prior uses of Word2vec in the studies of science, such as Tshitoyan et al. Nature 2019.

Repeating the link to the web app in the main text would be convenient.

Seeing Figure 6D and 6E, I would enjoy the authors showing or discussing more explicitly, whether textual differences increase with the number of revisions (and/or if there were some more complex changes such as reversions to earlier versions).

Reviewer #3:

This study asks an important question: (how) do preprints change between their initial release on a preprint server and their eventual publication in a peer-reviewed journal? While the analysis of the linguistic changes doesn't reveal anything particularly exciting (mostly typesetting and references to supplementary information included in response to reviewer requests), this is an incredibly useful result in demonstrating that preprints are typically of high quality, which has broad implications for how researchers and their work are assessed in career, funding, and publishing decisions. The authors have developed some very promising deliverables based on document embeddings that should be broadly applicable to readers, authors, journal editors, and other stakeholders navigating the complex landscape of preprinted and published literature.

Major Comments:

The method for discovering unannotated preprint-publication relationships is very neat, but I imagine it's rather unwieldy to match a novel publication against the full-text bioRxiv corpus in downstream applications (e.g., bioRxiv's automation)--could this be optimized by reducing the search space to preprints that share some or all of the same authors, within a reasonable date range, and/or only considering paper/preprint metadata (e.g., abstract, title, references)? Such an approach might also enable annotation of preprints that are eventually published as non-OA peer reviewed articles for which such metadata are available.

Section "Building Classifiers to Detect Linguistically Similar Journal Venues and Published Articles":

"Specific journals publish articles in a focused topic area, while others publish articles that cover many topics. Likewise, some journals have a publication rate of at most hundreds of papers per year, while others publish at a rate of at least ten thousand papers per year. Accounting for these characteristics, we designed two approaches - one centered on manuscripts and another centered on journals." << this could use some unpacking and/or reorganizing of details found later in this section--as I understand it, the variation in journals' topical breadth motivates the development of a manuscript-focused classifier (so that topically similar papers appearing in generalist journals do not get obscured) and the variation in journals' publication rates motivates a journal-focused classifier (so that high-output journals do not overwhelm more selective or less popular journals). 

I'm also curious how often these two classifiers agree--are the top matching papers typically found in the top matching journals? In cases where the two classifiers tend to disagree, are there any common characteristics of the preprints the application is trying to classify?

Minor Comments (by section):

Introduction:

The references of text mining on biomedical corpora should include Desai et al (2018) [https://www.biorxiv.org/content/10.1101/333922v1.abstract], which describes a similar recommendation engine.

Section "Comparing Corpora":

Inconsistent formatting of "spaCy"

Define "stopwords," since many readers may be unfamiliar with this term

Section "Constructing a Document Representation for Life Sciences Text":

This switches back to using "words" instead of "tokens" as in the previous section

Section "Measuring Time Duration for Preprint Publication Process":

Does this include the new preprint-publication pairs discovered in the previous section, or only those annotated in the data provided by the bioRxiv API?

Section "Preprints with more versions or more text changes took longer to publish":

Fig. 4: can the longer publication times for scicomm/education papers (Fig 4a) be explained by a tendency to go through more versions (Fig 4b)?

It might be worthwhile to explore what happens post-publication to papers that go through more preprint revisions and take longer to publish, as this could have practical implications for authors as they decide when/if to submit/revise their preprints. Do these papers ultimately receive more citations, end up in journals with higher impact factors, or receive more attention on social media?

Section "Preprints with similar document embeddings share publication venues":

From personal experience, converting bioRxiv PDFs to text sometimes introduces weird noise and artifacts. Since bioRxiv and medRxiv both offer full-text HTML for many (if not all?) articles, is it possible to modify the application to use this cleaner data source?

Section "Contextualizing the Preprints in Motion Collection":

Figure description for Fig 6E is mislabeled as D

There are several casually/awkwardly-worded or grammatically incorrect sentences throughout that could use some finesse:

Introduction:

"We hypothesize that preprints and biomedical text are pretty similar…"

Measuring Time Duration for Preprint Publication Process:

"Preprints that are published can take varying amounts of time to be published."

"We accomplish this by first randomly sampled with replacement a pair of preprints…"

Building Classifiers to Detect Linguistically Similar Journal Venues and Published Articles:

"Preprints are more likely to be published in journals that contained similar content to work in question."

Web Application for Discovering Similar Preprints and Journals:

"The application downloads a pdf version of any preprint hosted on the bioRxiv or medRxiv server uses PyMuPDF to extract text from the downloaded pdf and feeds the extracted text into our CBOW model to construct a document embedding representation."

Preprints with more versions or more text changes took longer to publish:

"Each new version adds additional 51 days before a preprint is published."

---

## [Editor Report · Decision Letter 2]

22 Oct 2021

Dear Dr Greene,

Thank you for submitting your revised Meta-Research Article entitled "Linguistic analysis of the bioRxiv preprint landscape" for publication in PLOS Biology. The Academic Editor and I have now assessed your revisions and responses to the reviewers.

Based on this assessment, we will probably accept this manuscript for publication, provided you satisfactorily address the following data and other policy-related requests.

IMPORTANT: Please address my Data Policy requests below; specifically, please supply numerical values underlying Figs 1ABCDE, 2ABC, 3ABC, 5ABCDE, S1ABCDE, S2, S3AB, S4AB, S5, S6AB, and cite the location of the data clearly in each relevant main and supplementary Fig legend. We note your comments about the raw NYTAC data, and we believe that this complies with the PLOS policy exemption for 3rd party data. However, we will need the numerical values plotted in the figs to be made available, perhaps as part of your Github deposition.

We expect to receive your revised manuscript within two weeks. 

*Published Peer Review History*

*Early Version*

Sincerely,

Roli Roberts

Senior Editor,

rroberts@plos.org,

PLOS Biology

DATA POLICY:

Regardless of the method selected, please ensure that you provide the individual numerical values that underlie the summary data displayed in the following figure panels as they are essential for readers to assess your analysis and to reproduce it: Figs 1ABCDE, 2ABC, 3ABC, 5ABCDE, S1ABCDE, S2, S3AB, S4AB, S5, S6AB. NOTE: the numerical data provided should include all replicates AND the way in which the plotted mean and errors were derived (it should not present only the mean/average values).

DATA NOT SHOWN?

---

## [Editor Report · Decision Letter 3]

5 Nov 2021

Dear Dr Greene,

On behalf of my colleagues and the Academic Editor, Ulrich Dirnagl, I'm pleased to say that we can in principle accept your Meta-Research Article "Examining linguistic shifts between preprints and publications" for publication in PLOS Biology, provided you address any remaining formatting and reporting issues. These will be detailed in an email that will follow this letter and that you will usually receive within 2-3 business days, during which time no action is required from you. Please note that we will not be able to formally accept your manuscript and schedule it for publication until you have any requested changes.

IMPORTANT: Note that as we try to avoid the use of punctuation in article titles, I've removed the initial phrase from your title.

PRESS: We frequently collaborate with press offices. If your institution or institutions have a press office, please notify them about your upcoming paper at this point, to enable them to help maximise its impact. If the press office is planning to promote your findings, we would be grateful if they could coordinate with biologypress@plos.org. If you have not yet opted out of the early version process, we ask that you notify us immediately of any press plans so that we may do so on your behalf.

Sincerely, 

Roli Roberts

Roland G Roberts, PhD 

Senior Editor 

PLOS Biology

rroberts@plos.org